# Role of IL-4 in bone marrow driven dysregulated angiogenesis and age-related macular degeneration

Takashi Baba[1]*, Dai Miyazaki[1]*, Kodai Inata[1], Ryu Uotani[1], Hitomi Miyake[1], Shin-ichi Sasaki[1], Yumiko Shimizu[1], Yoshitsugu Inoue[1], Kazuomi Nakamura[2]

[1]Division of Ophthalmology and Visual Science, Faculty of Medicine, Tottori University, Yonago, Japan; [2]Division of Pathological Biochemistry, Department of Biomedical Sciences, Faculty of Medicine, Tottori University, Tottori, Japan

**Abstract** Age-associated sterile inflammation can cause dysregulated choroidal neovascularization (CNV) as age-related macular degeneration (AMD). Intraocular fluid screening of 234 AMD patients identified high levels of IL-4. The purpose of this study was to determine the functional role of IL-4 in CNV formation using murine CNV model. Our results indicate that the IL-4/IL-4 receptors (IL4Rs) controlled tube formation and global proangiogenic responses of bone marrow cells. CCR2$^+$ bone marrow cells were recruited to form very early CNV lesions. IL-4 rapidly induces CCL2, which enhances recruitment of CCR2$^+$ bone marrow cells. This in vivo communication, like quorum-sensing, was followed by the induction of IL-4 by the bone marrow cells during the formation of mature CNVs. For CNV development, IL-4 in bone marrow cells are critically required, and IL-4 directly promotes CNV formation mainly by IL-4R. The IL-4/IL-4R$\alpha$ axis contributes to pathological angiogenesis through communications with bone marrow cells leading to retinal degeneration.

## Introduction

Age-related macular degeneration (AMD) is a neurodegenerative disorder which develops in elderly individuals and is a major cause of visual impairments in developed countries. In the early stages of AMD, lipoprotein deposits called drusen accumulate in the subretinal space between the photoreceptors and retinal pigment epithelium (RPE). Drusen are associated with the degeneration of the RPE which then leads to a dysfunction or loss of the photoreceptors. Choroidal (CNVs) develop in the subretinal space, and the CNVs lead to degeneration of the photoreceptor cells, infiltration by inflammatory cells, activation of microglia, and ganglion cell loss (*Beck et al., 2016*; *Copland et al., 2018*).

The pathology of AMD is coupled with senescence-associated para-inflammation, which is characterized by the secretion of IL-6, IL-8, CCL2, and CX3CL1 (*Sasaki et al., 2010*). Of these, CCL2 plays an important role in recruiting bone marrow cells, monocytes, and macrophages to the ocular neo-vascularizations. In this disease process, the bone marrow plays an important role by supplying new vascular endothelial cells and macrophages to the retina (*Gao et al., 2016*; *Zhou et al., 2017*). Thus, the bone marrow plays a key role in the repair of damaged tissues.

The M1 macrophages are functionally classified as pro-inflammatory, and the M2 macrophages are classified as anti-inflammatory, and both types are recruited to damaged tissues. The M2 macrophages are induced by IL-4, and it has been suggested that they have disease-regulating functions as opposed to the M1 macrophages (*Zhou et al., 2017*).

The concept of IL-4 as a regulatory and neuroprotective cytokine is supported by the findings in other neurodegenerative diseases including Alzheimer's disease (*Kiyota et al., 2010*) and

*For correspondence:
babatakashi8@gmail.com (TB);
miyazaki-ttr@umin.ac.jp (DM)

Parkinson's disease (*Chitnis and Weiner, 2017*; *Zhao et al., 2006*). Moreover, IL-4 is known to be a potent inhibitor of angiogenesis (*Haas et al., 2006*; *Volpert et al., 1998*), and thus may prevent pathological angiogenesis in eyes with AMD.

The purpose of this study was to determine whether bone marrow cells and IL-4 protect the photoreceptors from neurodegeneration, and whether they play regulatory roles in eyes with g AMD. To accomplish this, we first determined the concentration of IL-4 and other inflammatory cytokines in the aqueous humor of the eyes of AMD patients (*Sasaki et al., 2012*). We then determined whether IL-4 and bone marrow cells play roles in protecting the eye from abnormal angiogenesis. This was done by functional assays and global transcriptional profiling of bone marrow cells derived from endothelial progenitor cells (EPC).

## Results

### Increased levels of IL-4 in aqueous humor of eyes with AMD and clinical subtypes of AMD

We first examined the levels of IL-4 and related cytokines in the aqueous humor of human eyes with AMD. To accomplish this, aqueous humor was collected from the eyes of 234 patients with clinically-diagnosed AMD and impaired central vision and 104 normal subjects undergoing routine cataract surgery. The mean age of the patients with AMD was 74.1 ± 0.6 years, and it was 74.9 ± 1.0 years for the normal subjects. The results showed that the AMD patients had significantly higher levels of IL-4 in their aqueous than in normal subjects (*Table 1*, *Table 2*). In contrast, there was no significant elevation of IL-13.

We next examined whether the IL-4 levels were significantly associated with the different subtypes of AMD. The results showed that the level of IL-4 was significantly higher in the three clinical subtypes of AMD, for example, typical AMD, polypoidal choroidal vasculopathy (PCV), and retinal angiomatous proliferation (RAP). The degree of elevation of IL-4 (quintile) had the highest relative-risk ratio of 2.5 for RAP (p=0.001, logistic regression analysis after age adjustments), followed by 2.1 for typical AMD (p=0.000) and 1.7 for PCV (p=0.000) (*Table 2*).

### IL-4 induction in murine experimental choroidal neovascularization

The level of IL-4 expression was evaluated in a murine laser-induced CNV model to determine whether IL-4 is associated with subretinal neovascularization. First, we assessed whether the mRNA of IL-4 was induced in the CNV lesions. Our results showed that the mRNA of IL-4 was elevated and peaked at 3 days after the laser exposure and then decreased (*Figure 1a*). The mRNA of IL-4Rα also had similar induction kinetics. The mRNA of CCR2, a myeloid cell recruitment marker, was elevated, and the elevation preceded the mRNA of IL-4 induction by peaking at 1 day. The mRNA of CD11b gradually increased after the exposure.

To examine the spatial expression of IL-4, we examined the CNV lesions by immunohistochemistry. Three day after the laser exposure, the IL-4-expressing cells were observed along the margins of the lesions and were present more centrally on day 7 (*Figure 1b*). The IL-4-expressing cells were largely CD11b[+], and they were considered to be myeloid- or macrophage-lineage cells. The kinetics of Iba1-, CCL2-, and CD11b-positive cells after laser exposure was consistent with that of the mRNA induction (*Figure 1—figure supplement 1*).

We then examined which type of lineage cells can produce the CCL2 as an early recruitment signal for myeloid cells. Our results showed that the CCL2 was mainly associated with iba1-positive

**Table 1.** Increase of IL-4 concentration in aqueous humor of eyes with age-related macular degeneration.

| Cytokines (pg/ml) | Control (n = 104) | age-related macular degeneration (n = 234) | *P* value |
|---|---|---|---|
| IL-4 | 0.3 ± 0.1 | 0.9 ± 0.1 | p=0.0000 |
| IL-13 | 3.8 ± 0.7 | 5.2 ± 0.7 | NS |

Two-tailed t test; Mean ± standard error of the means (SEMs).

**Table 2.** Association of IL-4 concentration in aqueous humor with subtype of age-related macular degeneration.

|  | N | Relative risk ratio | | Relative risk ratio | |
|  |  | IL-4 (quintile) | *P* value | IL-13 (quintile) | *P* value |
| --- | --- | --- | --- | --- | --- |
| Control | 104 | - | - | - | - |
| Typical AMD | 33 | 2.11 ± 0.33 | 0.000 | 2.09 ± 0.39 | 0.000 |
| Polypoidal choroidal vasculopathy (PCV) | 78 | 1.70 ± 0.17 | 0.000 | 1.39 ± 0.15 | 0.002 |
| Retinal angiomatous proliferation (RAP) | 11 | 2.46 ± 0.69 | 0.001 | 1.48 ± 0.37 | 0.11 |

Multinomial logistic regression analysis after age adjustment; Mean ± standard error of the means (SEMs).

retinal microglial cells (*Figure 1c*). The microglial cells migrated to surface of the CNV (*Figure 1—video 1*). This indicated that they were the initial stimulators. Thus, IL-4 expressions followed by myeloid cell activation were early events acting at the inductive phase of the CNV formation.

## Requirement of IL-4 in inductive phase of choroidal neovascularization

The kinetic observations suggested that IL-4 appeared in the inductive phase of CNV formation. To determine whether IL-4 had inhibitory or stimulatory effects on the pathological angiogenesis, mice were laser-treated to induce the formation of CNVs, and IL-4 was injected intravenously on day 0 and day 3 during the inductive phase. The IL-4 significantly exacerbated the CNV formation in a dose dependent way (*Figure 2a*).

It is known that IL-4 generally signals through IL-4Rα which is also a ligand of IL-13. Therefore, we also tested whether IL-13 had any stimulatory effect on CNV formation. Our results showed that a systemic administration of IL-13 in the inductive phase had no significant effect on CNV formation.

To examine the role of IL-4 in the inductive phase of CNV development in more detail, laser-treated mice were injected intravenously with an anti-IL-4 antibody on day 0 and day 3 to try to inhibit the expression of IL-4 (*Figure 2b*). Consistent with the effects of IL-4 administration, an IL-4 blockade significantly reduced the size of the CNV. In contrast, a block of IL-13 by an antibody injection had no significant effect on the CNV formation.

We next evaluated the contribution of IL-4 to the formation of CNVs using *Il4*-deficient mice (*Figure 2c*). Consistent with the outcomes of the anti-IL-4 antibody exposure, *Il4*-deficient mice were significantly impaired in CNV formation which supports our finding that IL-4 is involved in the inductive phase of CNV. To confirm that IL-4 contributed to the CNV formation with canonical signaling by IL-4Rα in more detail, *Il4ra*-deficient mice were tested for CNV formation. *Il4ra*-deficiency impaired the CNV formation significantly (*Figure 2d*).

Next, we evaluated role of the IL-13 receptors as alternative receptors of IL-4. IL-13R is composed of IL-4Rα and IL-13Rα1. When IL-13Rα1-deficient mice were tested for CNV formation, no significant impairment was observed.

## Incorporation of circulating angiogenic cells and bone marrow-derived cells into CNV lesion

It has been shown that bone marrow cells are recruited into the CNV lesions during the inductive phase of CNV formation (*Gao et al., 2016*). Therefore, we examined the roles played by bone marrow-derived cells in CNV formation using bone marrow chimeric mice. Bone marrow chimeric mice were generated by reconstitution with a GFP transgenic mice-bone marrow, and they were evaluated for laser-induced CNV formation (*Figure 2e*).

Recruitment of GFP[+] bone marrow-derived cells (green) peaked at 3 days after irradiation. Thus, bone marrow-derived cell recruitment also contributed to the inductive phase process. Bone marrow-derived cells in this phase were CD11b[+] lineage, and they were positive for CCL2. These bone marrow cells did not express iba1 and were morphologically distinct from microglial cells. This suggested that these cells will amplify the recruitment of CCR2[+] lineage cells (*Figure 2—figure supplement 1*). Seven days after irradiation, bone marrow-derived cells were incorporated into structures formed by CD31[+] endothelial cells (*Figure 2—video 1*).

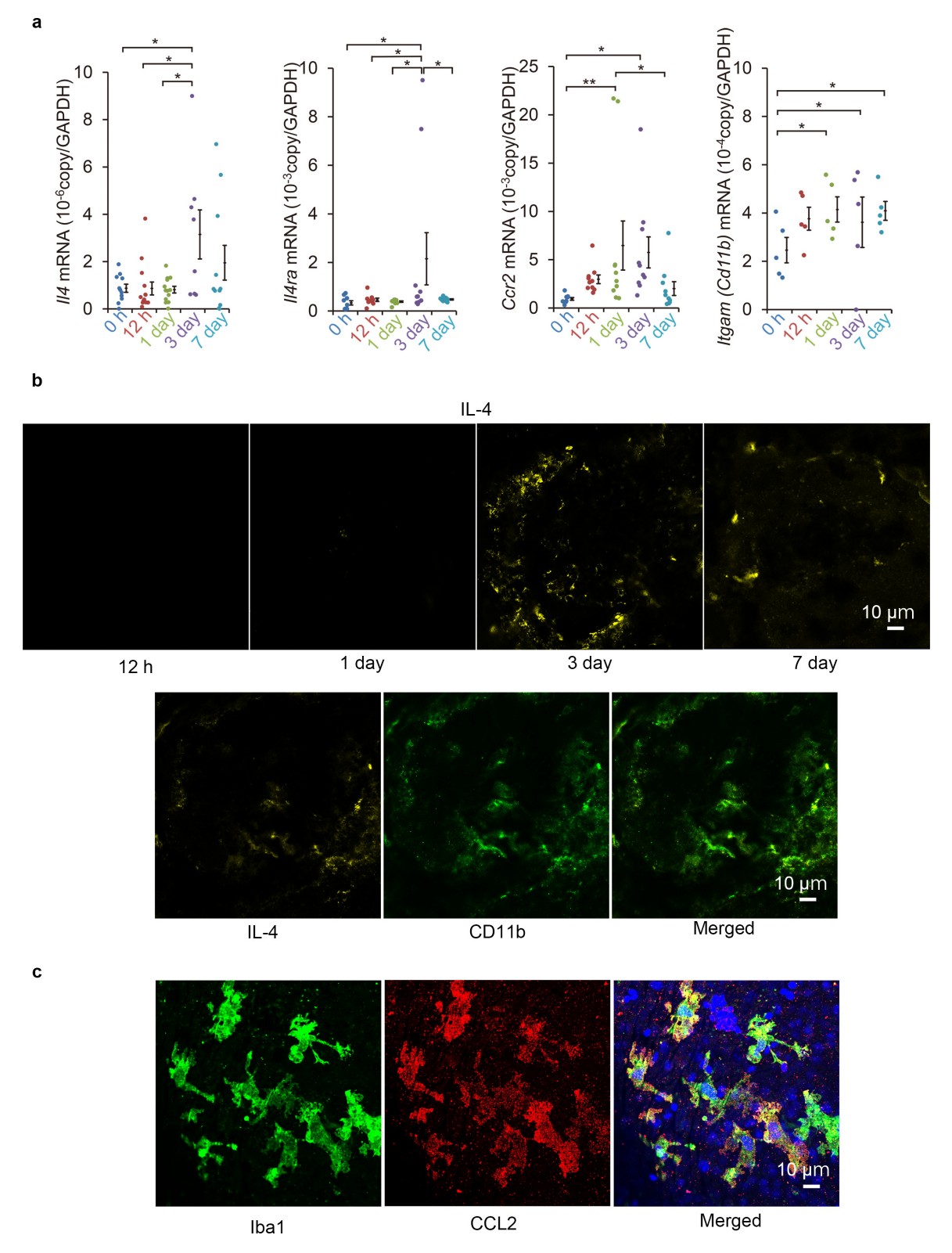

**Figure 1.** Induction of *Il4* and *Ccl2* in laser-exposed retinas and choroids of mice.  (a) Induction kinetics of the mRNAs of IL-4, IL-4Rα, CCR2, and CD11b. The induction of *Ccl2* peaked at 1 day after the exposure followed by the peak induction of *Il4* and *Il4ra*. (n = 4–14 eyes/group). (b) Kinetics of IL-4-expressing cells by immunohistochemical analyses. IL-4 expressing cells (yellow) accumulated along the margin of laser treated area at 3 days and then move inwards. The IL-4-expressing cells (yellow) were mainly CD11b positive (green) in laser treated areas at 3 days. (c) Localization of CCL2

*Figure 1 continued on next page*

*Figure 1 continued*

expression in retinal tissue by immunohistochemistry. CCL2 induction is observed at 1 day after treatment, and CCL2-positive cells (red) are colocalized with the iba1-positive microglial cells (green). The nuclei were stained by TO-PRO-3 iodide (blue). Scale 10 μm. *p<0.05, **p<0.01. ANOVA with post hoc test and linear mixed-effects regression analysis.

The online version of this article includes the following video, source data, and figure supplement(s) for figure 1:

**Source data 1.** Induction kinetics of the mRNAs of IL-4, IL-4Rα, CCR2, and CD11b.

**Figure supplement 1.** Kinetics of IL-4-expressing cells by immunohistochemical analyses in retinal flat mount.

**Figure 1—video 1.** Localization of bone marrow-derived cells and microglial cells in the retinal tissue of GFP bone marrow chimeric mouse in 3D rendering at 3 days after laser irradiation.

https://elifesciences.org/articles/54257#fig1video1

Two weeks after laser exposure, CNVs were formed as clusters of isolectin-positive vascular endothelial cells (red; *Figure 2e*). In the CNV lesion, bone marrow-derived cells (green) were localized to isolectin-positive vascular endothelial cells and CD31$^+$ endothelial cells. The co-localization of the marrow-derived cells with CD31$^+$ endothelial cells indicated that the bone marrow-derived cells may be able to differentiate into endothelial cells. IL-4-positive cells (yellow) were distributed at the margins of the CNVs and precisely matched the bone marrow-derived cells (green). The IL-4Rα-positive cells (cyan) in the CNV, partly overlapped the bone marrow-derived cells.

## Profiles of angiogenic mRNAs of endothelial progenitor cells

These findings suggested that the IL-4 from bone marrow-derived vascular endothelial cells played disease-promoting roles in CNV formation, and they were not anti-angiogenic. To confirm this, we examined how IL-4 affected the differentiation of vascular endothelial progenitor cells (EPC) from bone marrow cells. To do this, bone marrow cells were cultured for differentiation to late EPCs for 2 weeks and exposed to IL-4. We then screened for the induction of angiogenesis-related mRNAs, including *Ccl2*, *Vegf*, VEGF receptors (*Kdr*, *Flt4*), angiopoietin-1 (*Angpt1*), endothelin receptor (*Ednrb*), thrombin receptors (*F2r*, *F2rl1*), P-selectin (*Selp*), and vascular endothelial cadherin (*Cdh5*) (*Figure 3a*, *Figure 3—figure supplement 1*, *Figure 3—figure supplement 2*). Of these, *Ccl2* and *Flt1* were significantly induced in a dose dependent manner after IL-4 exposure. *Kdr* and *Flt4* were not induced.

We also examined whether mature vascular endothelial cells can induce comparable transcriptional responses. When retinal microvascular cells were tested for their effect on IL-4 by real-time reverse transcription PCR (RT-PCR), IL-4 was found to stimulate the induction of *Ccl2/Flt1* (*Figure 3b*).

An upregulation of the translation of CCL2 and VEGFR-1 in EPCs was confirmed by ELISA. IL-4-exposed EPCs had a significant increase in the secretion of CCL2 (p=0.000) and VEGFR-1 (p=0.000) after 24 hr exposure to IL-4 (*Figure 3—figure supplement 2*).

We next examined how IL-4Rα and IL-13Rα1 contributed to the induction of *Ccl2* and *Flt1* in EPCs by IL-4, IL-13, and VEGF. Both IL-4 and IL-13 significantly induced *Ccl2* and *Flt1* in EPCs (*Figure 3*). However, VEGF did not significantly induce *Ccl2* and *Flt1*. When IL-4Rα was inhibited by an anti-IL-4Rα antibody, IL-4 failed to stimulate the EPCs from inducing the expression of *Ccl2* and *Flt1* (*Figure 3a*).

To determine the contribution of IL-4Rα to CNV formation, we examined the effect of *Il4ra* deficiency. EPCs from *Il4ra*-deficient mice did not induce *Ccl2* and *Flt1* in response to IL-4 or IL-13 (*Figure 3c*).

We also examined the contribution of IL-13Rα1 to the formation of CNVs. When *Il13ra1*-deficient EPC mice were stimulated by IL-4, *Ccl2* and *Flt1* were still induced (*Figure 3d*) indicating that IL-13Rα1 was not necessary for IL-4 stimulation. When *Il13ra1*-deficient EPC mice were stimulated with IL-13, *Ccl2* and *Flt1* were not induced. Collectively, these findings indicate that IL-4Rα is the major receptor for IL-4 to induce the expression of CCL2 and VEGFR-1, and the IL-13Rα1 can substitute for their induction mainly through IL-13.

We next examined whether the *Ccl2* and *Flt1* induction by IL-4 required intrinsic IL-4-mediated differentiation. The results indicated that the EPCs of *Il4*-deificient mice still induced IL-4 and the IL-13-mediated *Ccl2* and *Flt1* induction (*Figure 3e*).

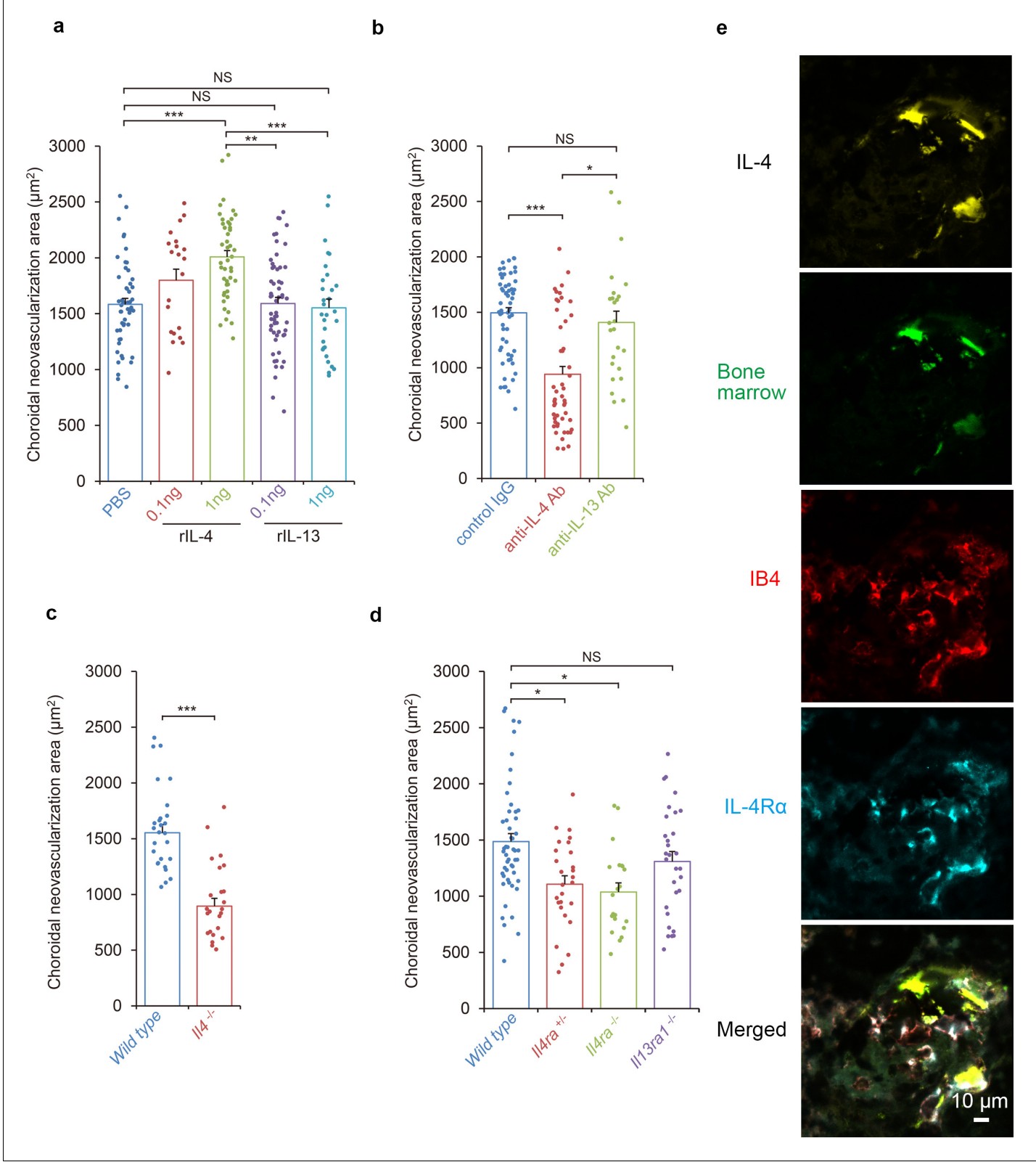

**Figure 2.** Requirements of IL-4/IL-4Rα in the inductive phase of choroidal neovascularization (CNV). (a) Effect of systemic administration of recombinant murine IL-4 (rIL-4) or recombinant IL-13 (rIL-13) in the inductive phase in a CNV model. (n = 7–19 eyes/group). (b) Inhibitory effect of systemic administration of anti-IL-4 antibody in the inductive phase of CNV formation. (n = 7–20 eyes/group). (c) Impaired CNV development in *Il4* deficient mice. CNV development is significantly impaired in *Il4*[-/-] mice compared to *wild type*. (n = 8–9 eyes/group). (d) Impaired CNV development by IL-4

*Figure 2 continued on next page*

*Figure 2 continued*

receptors deficiency. CNV development is significantly impaired in *Il4ra$^{-/-}$* and *Il4ra$^{+/-}$* mice compared to *wild type*. This impairment is more marked in the homozygotes. CNV development is not impaired for *Il13ra1$^{-/-}$* mice. (n = 7–17 eyes/group) (e) Bone marrow chimeric mice reconstituted with *GFP* transgenic bone marrow cells that were exposed to laser to induce CNVs. The CNV lesions after 14 days were analyzed for lineage cell markers by immunohistochemistry. CNVs are formed as clusters of isolectin IB4-positive vascular endothelial cells (red). Bone marrow-derived cells (green) were co-localized with isolectin-positive vascular endothelial cells. IL-4 positive cells (yellow) are distributed at the margins of the CNVs and precisely match the location of the bone marrow-derived cells (green). IL-4Rα-positive cells (cyan) partly overlapped the bone marrow-derived cells, and precisely match the location of the vascular endothelial cells in the CNV lesion. *p<0.005, **p<0.001, ***p<0.0005. Nested ANOVA with post hoc test. Scale 10 μm.
The online version of this article includes the following video, source data, and figure supplement(s) for figure 2:

**Source data 1.** Requirements of IL-4/IL-4Rα in the inductive phase of CNV.
**Figure supplement 1.** Kinetics of IL-4, IL-4Rα, CCR2 and CD11b-expressing cells and GFP-positive bone marrow derived cells determined by immunohistochemical analyses.
**Figure 2—video 1.** Localization of bone marrow-derived cells and endothelial cells in the retinal tissue of GFP bone marrow chimeric mouse in a 3D rendering 7 days after laser irradiation.
https://elifesciences.org/articles/54257#fig2video1

## Tube formation by endothelial progenitor cells and vascular endothelial cells stimulated by IL-4

To confirm a vasculogenic property of IL-4, mature vascular endothelial cells were assessed for tube formation on Matrigel-coated plates (*Figure 4a*). When murine retinal microvascular endothelial cells were tested for tube formation by IL-4 or VEGF, both stimulated significant tube formation (*Figure 4a*). Anti-IL-4 and IL-4Rα antibodies abolished the IL-4-induced tube formation.

We next confirmed the effects of IL-4 using human retinal cells (*Figure 4a*). IL-4 exposure stimulated tube formation by human retinal vascular endothelial cells, and anti-IL-4 and IL-4Rα antibodies blocked this effect.

Next, EPCs were examined for IL-4-mediated tube formation. Murine bone marrow cells were cultured under conditions appropriate for the differentiation of EPCs and were tested for tube formation. For the *wild type* bone marrow cells, IL-4 significantly stimulated tube formation by the EPCs (*Figure 4bc*). This IL-4-induced tube formation was blocked when the bone marrow cells were deficient of *Il4ra*. However, the inhibition of the VEGF receptor tyrosine kinase or VEGF receptor 2 did not significantly inhibit tube formation (*Figure 4b*). This indicated that this IL-4 effect was independent of canonical VEGF signaling.

The IL-4-induced tube formation was blocked when the cells were deficient of *Il4* but not by *Il13ra1* deficiency (*Figure 4c*). IL-13 also stimulated tube formation by bone marrow-derived EPCs. This response was abolished when the bone marrow cells were deficient in *Il4* and *Il13ra1*. These findings further confirmed the roles of IL-4 as a differentiation and vasculogenic factor which signaled mainly through IL-4Rα. The results also indicated that IL-13Rα1 could serve as an alternative receptor.

## IL-4Rα-dependent transcriptional networks of tube forming endothelial progenitor cells from bone marrow cells

The results suggested that IL-4 also served as a differentiation factor for cells of endothelial lineage. To characterize the vasculogenic roles of IL-4 for bone marrow cells, the mRNA of tube-forming EPCs were extracted and examined by network analysis. Analysis of the functions of IL-4-stimulated EPCs indicated significant association with the homing of the cells (Z score = 2.798, p=8.3 × 10$^{-5}$), angiogenesis (Z score = 2.781, p=1.8 × 10$^{-4}$), activation of macrophages (Z score = 2.731, p=1.0 × 10$^{-5}$), and recruitment of myeloid cells (Z score = 2.606, p=2.8 × 10$^{-10}$) (*Figure 4—figure supplement 1*).

## Requirements of IL-4 in bone marrow-mediated choroidal neovascularization

The results suggested that the IL-4 and IL-4Rα interactions contributed to the pathological angiogenesis of bone marrow-derived EPCs. To examine how bone marrow-derived cells contributed to IL-4-stimulated CNV formation, bone marrow chimeric mice were constructed on a *wild type* background of *Il4-* or *Il4ra-*deficient mice (*Figure 5*).

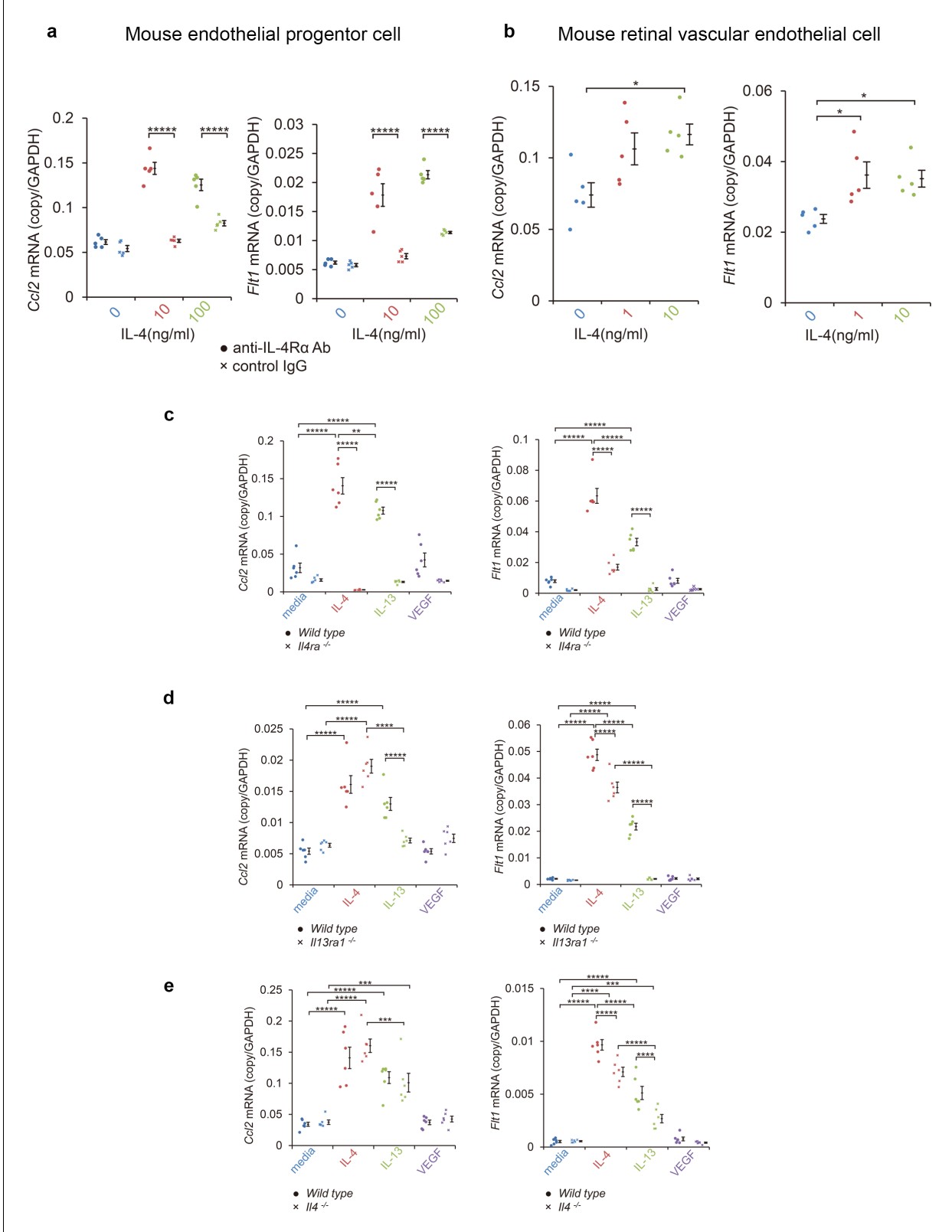

**Figure 3.** Induction of *Ccl2* and *Flt1* in bone marrow-derived endothelial progenitor cells (EPC) and retinal vascular endothelial cells by IL-4. (a) Induction of *Ccl2* and *Flt1* in bone marrow-derived endothelial progenitor cells by murine IL-4. IL-4 stimulated bone marrow-derived EPCs induced *Ccl2* and *Flt1* in a dose dependent manner. This induction is abolished by anti-IL-4Rα antibody. (n = 5/group). (b) Induction of *Ccl2* and *Flt1* in retinal vascular endothelial cells by IL-4. IL-4 stimulated vascular endothelial cells to express *Ccl2* and *Flt1* in a dose dependent manner. (n = 5/group). (c) *Figure 3 continued on next page*

*Figure 3 continued*

Inhibition of IL-4/IL-13-mediated *Ccl2* and *Flt1* induction in EPCs by *Il4ra* deficiency (n = 6/group). IL-4 and IL-13 exposure induced *Ccl2* and *Flt1* in EPCs. This induction is not present in the EPCs of *Il4ra$^{-/-}$* mice. (d) Inhibition of IL-13-mediated *Ccl2* and *Flt1* induction in EPCs by *Il13ra1* deficiency (n = 6/group). The IL-13-induced the expression of *Ccl2* and *Flt1* is significantly reduced in *Il13ra1$^{-/-}$* EPCs of mice. IL-4-induced *Ccl2* and *Flt1* mRNA is not affected in *Il13ra1$^{-/-}$* EPCs of mice. (e) EPCs of *Il4$^{-/-}$* mice respond to induce *Ccl2/Flt1* mRNA by IL-4/IL-13 exposure. (n = 6/group). *p<0.05, **p<0.01, ***p<0.005, ****p<0.0001, *****p<0.0005. ANOVA with post hoc test.

The online version of this article includes the following source data and figure supplement(s) for figure 3:

**Source data 1.** Induction of *Ccl2* and *Flt1* in bone marrow-derived EPC and retinal vascular endothelial cells by IL-4.
**Figure supplement 1.** Profile of angiogenic mRNAs of bone marrow-derived endothelial progenitor cells (EPCs) after IL-4 exposure.
**Figure supplement 1—source data 1.** Profile of angiogenic mRNAs of bone marrow-derived EPCs after IL-4 exposure.
**Figure supplement 2.** The CCL2 and VEGFR-1 protein levels in bone marrow-derived EPCs after IL-4 exposure.
**Figure supplement 2—source data 1.** The CCL2 and VEGFR-1 protein levels in bone marrow-derived EPCs after IL-4 exposure.

The *Il4*-deficient mice with *Il4$^{-/-}$* bone marrow developed the smallest size CNVs of all the chimeric mice. This impaired CNV formation was completely restored by the transplantation of bone marrow cells from *wild type* mice. This indicated the crucial role played by IL-4 in bone marrow cells. In contrast, *wild type* mice with *Il4$^{-/-}$* bone marrow were still impaired for CNV formation which indicated that the host resident cell-derived IL-4 is limited in this activity (*Figure 5a*).

Consistent with the results shown in *Figure 2*, *Il4ra*-deficient mice with *Il4ra$^{-/-}$* bone marrow cells were impaired in the formation of CNVs. This impairment was restored by the transplantation of *wild type* bone marrow cells (*Figure 5a*).

We next examined whether IL-4 secreting cells were recruited from bone marrow or were derived from the host. To do this, we conducted immunohistochemical analyses of *Il4*-deficient mice reconstituted with *wild type* bone marrow cells. The results showed IL-4- and IL-4Rα-positive cells were present in the CNV lesion (*Figure 5b*; *Figure 5—figure supplement 1*). The IL-4-positive cells precisely matched the bone marrow-derived cells (green). In contrast, the IL-4Rα positivity partly overlapped with that of the bone marrow-derived cells. This indicated that bone marrow-derived cells are the major producer of IL-4, and bone marrow-derived cells and resident cells in the CNV via IL-4Rα recognized their signals.

To summarize, interactions of IL-4/IL-4Rα interactions with bone marrow cells are required for pathological CNV formation.

## Discussion

Our results showed that IL-4 played a crucial role in the pathogenesis of CNVs by directing the migration and activating the angiogenic bone marrow cells. IL-4 is the canonical Th2 cytokine and is secreted by an array of inflammatory cells including macrophages, monocytes, and activated retinal pigment epithelial cells (*Leung et al., 2009*). IL-4 is also recognized as a neuroprotective cytokine, and its action is not limited to the retina (*Adão-Novaes et al., 2009*). In axotomized retinas, the retinal ganglion cells are severely damaged by nitric oxide synthesis by activated glial cells. IL-4 significantly increases the survival of retinal ganglion cells, and prevents neurodegeneration caused by glial cell activation (*Koeberle et al., 2004*). In the thapsigargin-induced rod photoreceptor cell death model, IL-4 can completely block the death of the photoreceptors (*Adão-Novaes et al., 2009*). During the differentiation of the retina, IL-4 modulates the proliferation of the retinal cells and promotes photoreceptor differentiation (*da Silva et al., 2008*). In addition, a number of studies have shown that IL-4 can inhibit the death of photoreceptors and RGCs.

IL-4 is a multifaceted cytokine and is known to have anti-angiogenic capabilities. IL-4 inhibits tumor growth by inhibiting angiogenesis (*Volpert et al., 1998*) and also blocks corneal neovascularization induced by basic fibroblast growth factor. Thus, IL-4 can function as an anti-inflammatory cytokine and prevent neuronal death and angiogenesis. However, such properties of IL-4 appear to be context dependent.

We found that the IL-4 level was significantly elevated in the aqueous humor of patients with AMD (*Table 1*, *Table 2*; *Sasaki et al., 2012*). Together with this, our analyses of the bone marrow cells and chimeric mice supports the idea that elevations of IL-4- and IL-4 receptor-bearing cells are associated with the development of abnormal vessels in the lesions of eyes with AMD.

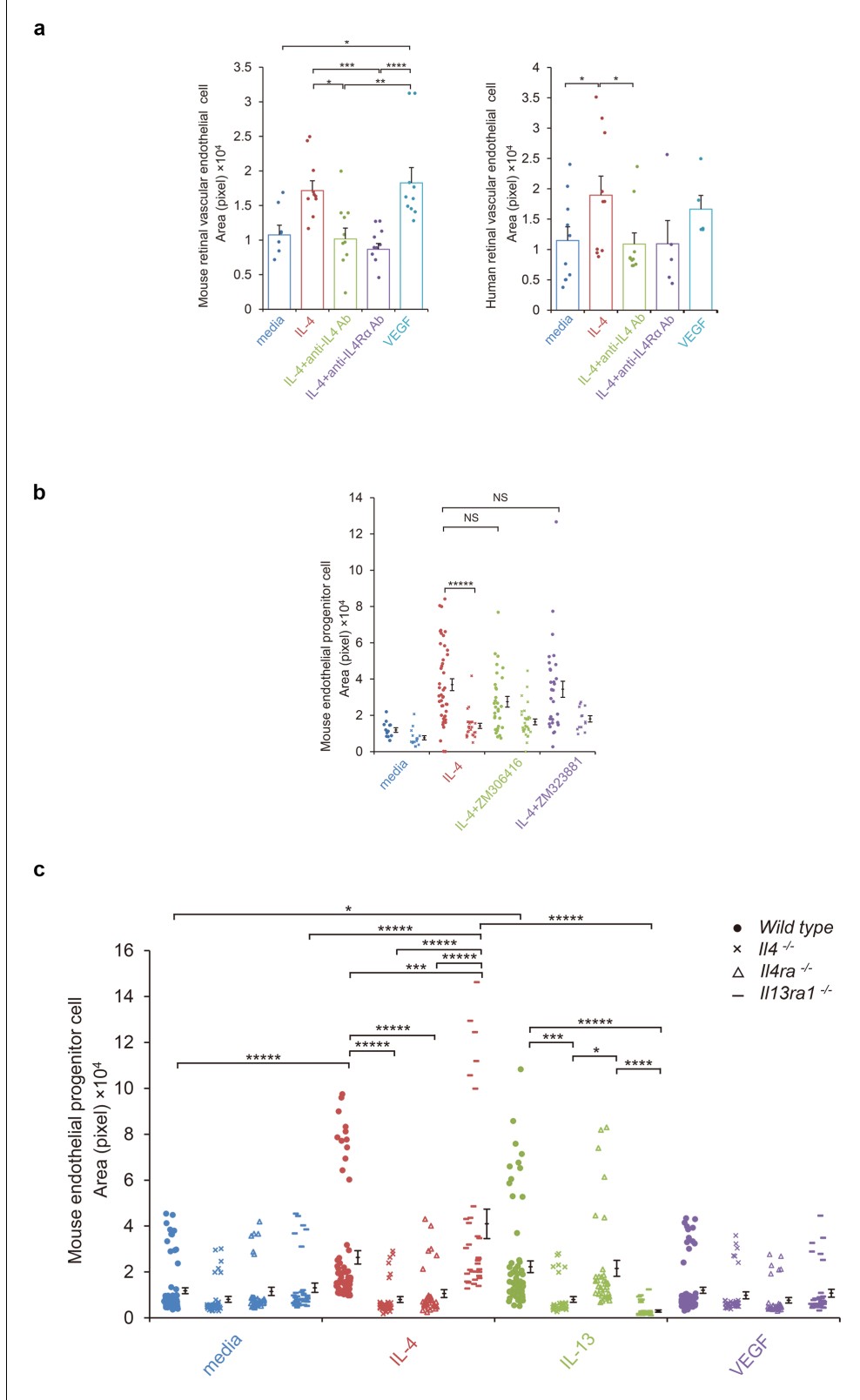

**Figure 4.** IL-4-induced tube formation in endothelial progenitor cells (EPCs) and retinal vascular endothelial cells. (**a**) IL-4-induced tube formation of retinal vascular endothelial cells. Human and murine IL-4 exposure (10 ng/ml) significantly stimulated tube formation of human and murine retinal microvascular endothelial cells in vitro, respectively. Anti-IL-4 or IL-4Rα antibodies abolished the IL-4 induced-tube formation. VEGF exposure (10 ng/ml) also stimulated tube formation. (n = 7–10/group). (**b**) IL-4-induced tube formation in bone marrow-derived EPCs. IL-4 exposure (10 ng/ml)

*Figure 4 continued on next page*

*Figure 4 continued*

significantly stimulated tube formation of EPCs. The IL-4-induced tube formation was significantly reduced in EPCs from *Il4ra*[-/-] mice but was not affected by inhibition of VEGF receptor tyrosine kinase (ZM 306416) or VEGFR-2 (ZM 323881) (n = 13–45/group). (c) Requirements of IL-4 for tube formation response of EPCs. IL-4 (10 ng/ml) and IL-13 (10 ng/ml) induced tube formation of bone marrow-derived EPCs. These actions were abolished in the EPCs from *Il4*[-/-] bone marrow cells. EPCs from *Il4ra*[-/-] mice did not respond to IL-4, however they responded to IL-13 by tube formation. EPCs from *Il13ra1*[-/-] mice did not respond to IL-13 but responded to IL-4 by tube formation. (n = 35–72/group). *p<0.05, **p<0.01, ***p<0.005, ****p<0.001, *****p<0.0005. ANOVA with post hoc test and linear mixed-effects regression analysis.

The online version of this article includes the following source data and figure supplement(s) for figure 4:

**Source data 1.** IL-4-induced tube formation inEPCs and retinal vascular endothelial cells.
**Figure supplement 1.** IL-4Rα-mediated transcriptional networks of bone marrow-derived EPCs.
**Figure supplement 1—source data 1.** IL-4Rα-mediated transcriptional networks of bone marrow-derived EPCs.

Although the IL-4/IL-4Rα axis appears neuroprotective in the retina, retinal injury requires a recruitment or activation of cells with regenerative properties for its repair. The bone marrow cells are a major supplier of mesenchymal stem cells and hematopoietic stem cells. However, abnormally activated bone marrow cells by IL-4 in the retina promote pathological angiogenic responses.

To explain the dysregulated repair process after organ damage, the concept of cell level and organ level quorum sensing has been recently proposed (*Antonioli et al., 2018*). Quorum sensing was originally proposed as a phenomenon of bacterial cells, and it was described as a mechanism that senses the environment and integrity of a population of cells. Hair follicle injury sensed by a macrophage-mediated circuit via CCL2 is a well-known example of quorum sensing at the organ level (*Chen et al., 2015*; *Feng et al., 2017*). The quorum sensing circuit mediated by microglia-derived CCL2 also appears to operate in new vessel formation in the retina together with a late IL-4 modulator.

We observed the presence of CD11b in the CNV lesions at 12 hr which would indicate that mono-cyte/macrophage cells had arrived soon after the beginning of the CNV. In contrast, the induction of IL-4 and IL-4 receptors was delayed and peaked at 3 days (*Figure 1*). CCL2 recruits circulating mono-nuclear cells from the bone marrow, and the retinal pigment epithelial cells and microglial cells are the major sources of the CCL2 cells in the retina (*Feng et al., 2017*). Thus, the monocyte/macro-phage recruiting signal, including CCL2, appears to be the first signal in the formation of CNVs.

It was reported that CCL2 is involved in the formation of retinal neovascularization (*Sennlaub et al., 2013*; *Yoshida et al., 2003*). However, the CCL2/CCR2 signals recruit a heterogeneous collection of monocyte/macrophage lineage cells and presumably do not determine their fate (*Grochot-Przeczek et al., 2013*; *Pearson, 2010*).

Bone marrow-derived cells and macrophages are critical contributors to retinal and choroidal neovascularization (*Gao et al., 2016*; *Zhou et al., 2017*). Higher levels of M1 than M2 type mRNAs were observed in advanced stage AMD patients (*Cao et al., 2011*). The M1 macrophages counteract the M2 type by the secretion of interferon-γ. Based on this, the M2 macrophages were considered to play a regulatory role in CNV formation. Consistent with this, Wu et al showed that the M2 type cytokine, IL-4, and conditioned macrophages become the regulatory phenotype to suppress the disease processes (*Wu et al., 2015*). However, the M1 types of IFNAR1+ macrophages have been reported to be protective in laser-induced CNV (*Lückoff et al., 2016*). Thus, the M1/M2 paradigm does not clearly explain how abnormal vessels are formed in AMD lesions or the CNVs of the model mice.

Wu et al also showed a contradictory role of IL-4 for CNV formation (*Wu et al., 2015*). This was shown using vitreous injection of IL-4 at very high concentration (600 ng/ml). This may cause toxic damage to endothelial cells or recruited cells which may not reflect physiological role of IL-4.

In the retina, vascular repair and neovascularization are performed largely by circulating EPCs because mature vascular endothelial cells for proliferation are limited (*Caballero et al., 2007*; *Grant et al., 2002*). EPCs are present in the bone marrow or are peripheral blood mononuclear cells (*Rohde et al., 2006*; *Schatteman and Awad, 2003*; *Schmeisser et al., 2001*), and they can differentiate into endothelial cells as late EPCs and be incorporated into the vasculature system. Alternatively, bone marrow cells will also differentiate into non-endothelial cell lineage and serve as providers of CNV-forming signals.

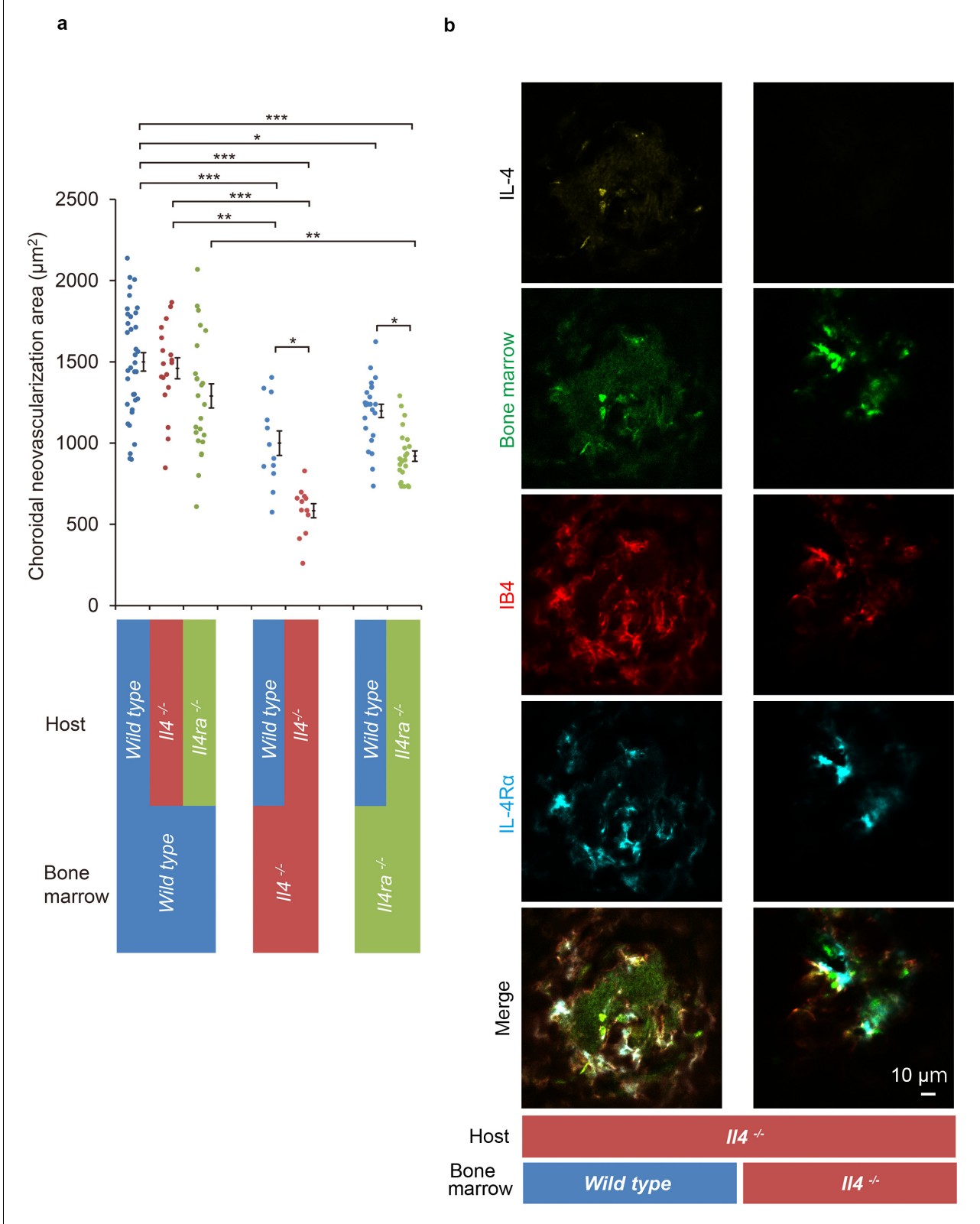

**Figure 5.** Role of IL-4/IL-4Rα on bone marrow-mediated choroidal neovascularization in bone marrow chimeric mice. (a) Requirement of IL-4 in bone marrow for choroidal neovascularization (CNV). Bone marrow chimeric mice were constructed on backgrounds of *wild type*, *Il4⁻/⁻*, or *Il4ra⁻/⁻* mice by transfer of *wild type Il4⁻/⁻*, or *Il4ra⁻/⁻* bone marrow cells. *Il4⁻/⁻* mice with *Il4⁻/⁻* bone marrow are most significantly impaired in CNV formation. This impairment of CNV formation is restored when bone marrow reconstituted with *wild type* bone marrow. *Il4ra⁻/⁻* mice with *Il4ra⁻/⁻* bone marrow cells are

*Figure 5 continued on next page*

*Figure 5 continued*

impaired in CNV formation. This impairment is partially restored when reconstituted with *wild type* bone marrow cells. (n = 4–12 eyes/group). Six of 10 *IL-4$^{-/-}$* bone marrow chimeric mice in each group did not survive through procedures and/or were euthanized. *p<0.05, **p<0.005, ***p<0.0005. Nested ANOVA with post hoc test. (b) Immunohistochemical analysis of CNV of the bone marrow chimeric mice on *Il4$^{-/-}$* background 14 days after laser treatment. Endothelial cells in the CNV were labeled with isolectin IB4 (red). In the *Il4$^{-/-}$* mice reconstituted with *wild type* bone marrow, the CNV lesion contained IL-4 (yellow) secreting bone marrow cells (green). The IL-4Rα-positive cells (cyan) partly overlapped with bone marrow-derived cells. Scale 10 μm.

The online version of this article includes the following source data and figure supplement(s) for figure 5:

**Source data 1.** Requirement of IL-4 in bone marrow for CNV.

**Figure supplement 1.** Immunohistochemical analysis of CNV of the bone marrow chimeric mice on *Il4ra$^{-/-}$* and *wild type* background 14 days after laser treatment.

In the inductive phase of CNVs, a mobilization of circulating angiogenic cells and monocyte/macrophage lineage cells begins by the early recruitment of IL-4 secreting CD11b$^+$ bone marrow cells (*Figure 1*). As a late phase phenomenon, the IL-4R$^+$ bone marrow-derived cells are incorporated into the CNV presumably as late EPCs or non-endothelial lineage cells, together with the resident cell-derived endothelial cells. We propose that the bone marrow-derived cells contribute to both phases using IL-4 as the fate determinant.

We also noted that the IL-4/IL-4Rα axis is involved in pathologic angiogenesis. For example, IL-4 induces proinflammatory phenotypes and causes vascular leakage or increased turnover of endothelial cells (*Kotowicz et al., 2004*; *Lee et al., 2010*). IL-4 stimulates human umbilical vascular endothelial cells (HUVEC) cells to induce proinflammatory cytokines including CCL2, VCAM-1, and IL-6, as a signature of global transcriptional responses (*Lee et al., 2004*). These responses appear to be a general characteristics of vascular endothelial cells including coronary arterial endothelial cells (*Skaria et al., 2016*).

To identify an endothelial lineage, we used CD31 or isolectin staining because CD31 is highly expressed on endothelial cells and is commonly used as an endothelial cell marker. However, CD31 can also be expressed on other lineage cells including T cells, B cells, dendritic cells (DCs; *Clement et al., 2014*), neutrophils, monocytes, and macrophage (*Merchand-Reyes et al., 2019*).

Our data support the idea that bone marrow-derived cells may be able to differentiate into endothelial cells in the CNV lesions. However, whether the endothelial differentiation is complete or bone marrow cells serve as immature or of different lineage was not definitively determined. Importantly, bone marrow-derived cells do play pivotal roles in the CNV formation.

In conclusion, damages of the retina and choroidal tissue release signals to the bone marrow to repair the vascular damage. This signal induces a recruitment of the bone marrow-derived cells for differentiation into or establishment of new vessels. Calling and/or fate determining signals are governed by IL-4. IL-4 may serve as a therapeutic target to treat this visual disorder.

## Materials and methods

**Key resources table**

| Reagent type (species) or resource | Designation | Source or reference | Identifiers | Additional information |
| --- | --- | --- | --- | --- |
| Genetic reagent (*M. musculus*) | *C57BL/6J (wt)* | PMID:15729571 | RRID:IMSR_JAX:000664 | |
| Genetic reagent (*M. musculus*) | *C57BL/6-Tg (CAG-EGFP)* | PMID:9175875 | RRID:IMSR_JAX:003291 | |
| Genetic reagent (*M. musculus*) | *C57BL/6-Il4$^{tm1Nnt}$/J* | PMID:8906833 | RRID:IMSR_JAX:002518 | |
| Genetic reagent (*M. musculus*) | *BALB/c-Il4ra$^{tm1Sz}$/J* | PMID:9380721 | RRID:IMSR_JAX:003514 | |

*Continued on next page*

*Continued*

| Reagent type (species) or resource | Designation | Source or reference | Identifiers | Additional information |
|---|---|---|---|---|
| Genetic reagent (*M. musculus*) | *Il13ra1*<sup>tm1Twy</sup> | PMID:18066066 | RRID:MGI:3772446 | Dr. Marc E Rothenberg, Cincinnati Children's Hospital Medical Center University of Cincinnati College of Medicine |
| Cell line (*M. musculus*) | *C57BL/6* Mouse Primary Retinal Microvascular Endothelial Cells | Cell Biologics | C57-6065 | |
| Cell line (*H. sapiens*) | Primary Human Retinal Microvascular Endothelial Cells | Cell Systems | ACBRI 181 | |
| Antibody | anti IL-4 (rat monoclonal) | Biolegend | Cat. #: 504108, RRID:AB_315322 | IHC(1:200) |
| Antibody | anti IL-4 (rabbit polyclonal) | abcam | Cat. #: ab9622, RRID:AB_308736 | IHC(1:200) |
| Antibody | anti Phospho-Tyr497 IL-4R/CD124 (rabbit polyclonal) | Assay Biotechnology Company | Cat. #: A1064, RRID:AB_10683571 | IHC(1:200) |
| | anti CD124 (rat monoclonal) | BD Pharmingen | Cat. #: 552288, RRID:AB_394356 | 0.1–10 ng/ mL TVI |
| Antibody | anti IL-13 receptor alpha 1 (rabbit polyclonal) | abcam | Cat. #: ab-79277, RRID:AB_1640587 | IHC(1:200) |
| | anti IL13 antibody (rabbit polyclonal) | GeneTex, Inc | Cat. #: GTX59763, | 0.1–10 ng/ mL TVI |
| Antibody | anti CD11b (rat monoclonal) | eBioscience | Cat. #: 14-0112-82, RRID:AB_467108 | IHC(1:200) |
| Antibody | anti CD11b (rat monoclonal) Alexa Fluor 594 | Biolegend | Cat. #: 101254, RRID:AB_2563231 | IHC(1:100) |
| Antibody | CD11b (M1/70) (rat monoclonal) FITC | eBioscience | Cat. #: 11-0112-41, RRID:AB_11042156 | IHC(1:100) |
| Antibody | anti CCR2 (rabbit polyclonal) DyLight 550 | Novus Biologicals | Cat. #: NBP1-48338R | IHC(1:100) |
| Antibody | anti MCP-1 (hamster monoclonal) | Biolegend | Cat. #: 505906, RRID:AB_2071552 | IHC(1:200) |
| Antibody | anti Iba1 (rabbit polyclonal) | FUJIFILM Wako Pure Chemical Corporation | Cat. # 019–19741, RRID:AB_839504 | IHC(1:200) |
| Antibody | anti CD31 (rabbit polyclomal) | abcam | Cat. #: ab28364, RRID:AB_726362 | IHC(1:200) |
| Antibody | anti-mouse CD31(rat monoclonal) Alexa Fluor 647 | Biolegend | Cat. #: 102516, RRID:AB_2161029 | IHC(1:100) |
| Antibody | anti rat IgG (goat polyclonal) Brilliant Violet 421 | Biolegend | Cat. #: 405414, RRID:AB_10900808 | IHC(1:100) |
| Antibody | anti rabbit IgG (goat polyclonal) DyLight 488 | Vector Laboratories | Cat. #: DI-1488, RRID:AB_2336402 | IHC(1:100) |
| Antibody | anti rabbit IgG (donkey polyclonal) Alexa Fluor 555 | Biolegend | Cat. #: 406412, RRID:AB_2563181 | IHC(1:100) |

*Continued on next page*

Continued

| Reagent type (species) or resource | Designation | Source or reference | Identifiers | Additional information |
|---|---|---|---|---|
| Antibody | anti rabbit IgG (goat polyclonal) HiLyte Fluor 555 | AnaSpec | Cat. #: AS-61056–05 H555 | IHC(1:100) |
| Antibody | anti hamster IgG (goat polyclonal) DyLight 594 | Biolegend | Cat. #: 405504, RRID:AB_1575119 | |
| Antibody | anti rabbit IgG (goat polyclonal) PE | Santa Cruz Biotechnology | Cat. #: sc-3739, RRID:AB_649004 | IHC(1:100) |
| Antibody | anti rabbit IgG (donkey polyclonal) Alexa Fluor 647 | abcam | Cat. #: ab150075, RRID:AB_2752244 | IHC(1:100) |
| Antibody | anti rabbit IgG (donkey polyclonal) DyLight 649 | Biolegend | Cat. #: 406406, RRID:AB_1575135 | IHC(1:100) |
| Antibody | anti mouse IgG2A (rat monoclonal) | R and D | Cat. #: mab006, RRID:AB_357349 | 0.1–10 ng/ mL TVI |
| Antibody | anti mouse Fc gamma RII/RIII (CD32/CD16)(goat polyclonal) | R and D | Cat. #: AF1460-SP, Accession # P08101 | IHC (0.2 µg/mL) |
| Peptide, recombinant protein | recombinant murine IL-4 | R and D | Cat. #: 404 ML | |
| Peptide, recombinant protein | recombinant human IL-4 | Peprotec | Cat. #: AF-200–04 | |
| Peptide, recombinant protein | recombinant murine IL-13 | Peprotec | Cat. #: 210–13 | |
| Chemical compound, drug | ZM306416 hydrochloride | abcam | Cat. #: ab144576 | |
| Chemical compound, drug | ZM323881 hydrochloride | R and D | Cat. #: 2475/1 | |
| Chemical compound, drug | bovine serum albumin | Sigma-Aldrich | Cat. #: A2153 | |
| Chemical compound, drug | fetal bovine serum | Sigma-Aldrich | Cat. #: 12103C | |
| Chemical compound, drug | Medetomidine | Chemscene LLC | Cat. #: 86347-14-0 | |
| Chemical compound, drug | Butorphanol tartrate | FUJIFILM Wako Pure Chemical Corporation | Cat. #: 58786-99-5 | |
| Chemical compound, drug | Midazolam | FUJIFILM Wako Pure Chemical Corporation | Cat. #: 59467-70-8 | |
| Chemical compound, drug | Tropicamide, Phenylephrine Hydrochloride | Santen Pharmaceuitical Co., Ltd. | Cat. #: 1319810Q1053 | |

*Continued on next page*

Continued

| Reagent type (species) or resource | Designation | Source or reference | Identifiers | Additional information |
|---|---|---|---|---|
| Chemical compound, drug | Hydroxy methyl cellulose | SENJU Pharmaceutical Co.,Ltd | Cat.#: 131980AQ1038 | |
| Chemical compound, drug | RNAlater solution | Ambion | Cat. #: AM7021 | |
| Chemical compound, drug | Triton X-100 | Sigma-Aldrich | Cat. #: X100 100ml | |
| Chemical compound, drug | Tween 20 | Sigma-Aldrich | Cat. #: P1379-100ml | |
| Chemical compound, drug | Paraformadehyde | Electron Microscopy Science | Cat. #: 15710 | |
| Commercial assay or kit | ISOLECTIN B4 Fluorescein | Vector Laboratories | Cat. #: FL-1201, RRID:AB_2314663 | IHC(1:100) |
| Commercial assay or kit | ISOLECTIN B4 DyLight 594 | Vector Laboratories | Cat. #: FL-1207 | IHC(1:100) |
| Commercial assay or kit | DAPI | Roche Diagnostics | Cat. #: 10 236 276 001 | 1 µg/ml |
| Commercial assay or kit | TO-PRO-3 iodide | Molecular Probes, Inc | Cat. #: T-3605 | IHC(1:100) |
| Commercial assay or kit | VECTASHIELDAntifade Mounting Medium | Vector Laboratories | Cat. #: H-1000, RRID:AB_2336789 | |
| Commercial assay or kit | Fluorescence Mounting Medium | DAKO | Cat. #: 15710 | |
| Commercial assay or kit | SurePrint G3 Mouse GE 8 × 60K Microarray | Agilent Technologies | Cat. #: AGLMO002 | |
| Commercial assay or kit | QuantiTect Reverse Transcription Kit | Qiagen | Cat. #: 205311 | |
| Commercial assay or kit | QuantiTect SYBR Green PCR kit | Qiagen | Cat. #: 204143 | |
| Commercial assay or kit | ELISA kits | ThermoFisher Scientific | Cat. #: BMS6005, EMFLT1 | |
| Commercial assay or kit | PKH26 Red Fluorescent Cell Linker Kit for General Cell Membrane Labeling | Sigma-Aldrich | Cat. #: PKH26GL-1KT | |
| Commercial assay or kit | RNeasy Mini Kit | Qiagen | Cat. #: 74104 | |
| Commercial assay or kit | DMEM:F12 | ThermoFisher Scientific | Cat. #: 11330057 | |
| Commercial assay or kit | recombinant human GMCSF | Peprotec | Cat. #: AF-300–03 | |
| Commercial assay or kit | recombinant murine GMCSF | Peprotec | Cat. #: 315–03 | |
| Software, algorithm | Ingenuity Pathway Analysis, March 2020 | Qiagen | RRID:SCR_008653 | |
| Software, algorithm | Adobe Photoshop CS5, Version 12.0.5 | Adobe | RRID:SCR_014199 | |
| Software, algorithm | STATA 16.1 | StataCorp LLC | RRID:SCR_012763 | |
| Software, algorithm | GeneSpring Software | Agilent Technologies | RRID:SCR_009196 | |

## Patient selection and measurement of aqueous humor IL-4

The diagnosis of AMD and subtypes of AMD including PCV and RAP was made by the clinical characteristics. The presence of a CNV or retinal angiomatous proliferation (RAP) was determined by fluorescein angiography, indocyanine green angiography, and spectral domain optical coherence tomography (SD-OCT).

The inclusion criteria were the presence of active CNVs or RAP lesions determined by the angiographic images showing macular edema or subfoveal hemorrhages. Eyes with laser photocoagulation, photodynamic therapy, or intraocular surgery within the past 3 months were excluded. For the control groups, aqueous humor was collected from normal patients who were undergoing routine cataract surgery.

The levels of IL-4 and IL-13 in the aqueous humor samples were measured by commercial ELISA kits as described in detail (*Chono et al., 2018*; *Sasaki et al., 2012*).

## Animals

*Il4*-deficient mice, C57BL/6-*Il4*$^{tm1Nnt}$/J, *Il4ra*-deficient mice, and BALB/c-*Il4ra*$^{tm1Sz}$/J, were obtained from The Jackson Laboratory (Bar Harbor, ME). The *Il4ra*-deficient mice were backcrossed with C57BL/6 for 9 generations. The *Il13ra1*-deficient mice were obtained from the Regeneron Pharmaceuticals (Tarrytown, NY). *Green fluorescent protein* (*GFP*) transgenic C57BL/6 and *wild type* C57BL/6 mice were purchased from Japan SLC Inc (Shizuoka, Japan).

## Induction of choroidal neovascularization

Choroidal neovascularization (CNV) was induced by laser irradiation of the retina of mice, an established model for choroidal or retinal neovascular formation. This model has many characteristics of age-related macular degeneration. Mice were anesthetized and one eye was exposed to argon laser irradiation of 150 mW for 0.10 s. Three laser spots were created in each eye. The spot size was approximately 50 μm, and it was delivered with the Novus 2000 argon laser system (Coherent, Santa Clara, CA).

To analyze the CNVs, the laser-treated eyes were enucleated from euthanized mice 14 days after the photocoagulation. Choroidal sheets were isolated from the eyes and fixed in 4% paraformaldehyde at 4° C for 1 min. The choroidal sheets were stained with FITC or DyLight 594 conjugated Isolectin IB4 (Vector Laboratories, Peterborough, UK) and flat-mounted. The stained flat mounts sections were examined and photographed with a fluorescence stereo microscope (MZ-III, Leica Microsystems, Wetzlar, Germany). The isolectin IB4 reactive areas were analyzed as the CNV area by masked investigators to measure the CNV size.

## Immunohistochemistry of choroidal neovascularization

For the immunohistochemical analyses of the CNVs, isolated choroidal sheets with or without the retina were fixed in 4% paraformaldehyde and incubated with isolectin IB4 and primary antibodies including anti-IL-4 (11B11, Biolegend, San Diego, CA), anti-IL-4Rα (mIL4R-M1, BD Biosciences, Franklin Lakes, NJ), anti-Iba1 (FUJIFILM, Tokyo, Japan), anti-IL-13R (ab-79277, abcam, Cambridge, UK), anti-CD11b (M1/70, eBioscience, San Diego, CA), anti-CCL2 (2H5, Biolegend), anti-CCR2 (NBP1-48338R, Novus Biologicals, Centennial, CO), or anti-CD31 (390, Biolegend) with anti-Fc RII/RIII blocking antibody (R and D Systems, McKinley Place, NE) overnight at 4° C. Then, the choroidal sheets were rinsed and incubated with the secondary antibodies labeled with either Brilliant Violet 421, DyLight 488, Alexa Fluor 555, PE, Alexa Fluor 647, or DyLight 649, or control antibodies. DAPI and TO-PRO-3 iodide (T-3605, Molecular Probes, Eugene, OR) were used for nuclear staining. A confocal microscope (LSM730, Carl Zeiss, Oberkochen, Germany), or a fluorescence microscope (BZ-X800, Keyence, Osaka, Japan) was used for photographing the whole mounts.

## Intravenous injection of IL-4/IL-13 and blockade by antibody

After the laser irradiation, recombinant mouse IL-4 (R and D Systems, Minneapolis, MN), IL-13 (Peprotec, Rocky Hill, NJ), or vehicle was injected through a tail vein on days 0 and 3. To block the induction of IL-4 or IL-13, anti-IL4 antibody (50 μg/mouse, BioLegend), anti-IL13 antibody (50 μg/mouse, Gene Tex, Irvine, CA), or control IgG was injected through the tail vein on days 0 and 3.

## Generation of bone marrow chimeric mice

Bone marrow cells were collected from the femur and tibia as described in detail (*Wu et al., 2015*). Recipient mice were irradiated (600 rad × 2) with a MX-160Labo Irradiator (MediXtec, Chiba, Japan) and then injected with a bone marrow cell suspension ($1 \times 10^7$ cells) through the tail vein. The transplanted mice were allowed to recover for 5 weeks to reconstitute their myeloid cells. The reconstitution was confirmed by flow cytometry and the staining of the bone marrow or blood samples.

*GFP* transgenic mice-derived bone marrow or PKH26 (Sigma, Saint Louis, MO) labeling was used to examine the bone marrow cells. The stability of the PKH labeling was examined in the chimeric mice transplanted with PKH-labeled *GFP*-transgenic bone marrow. A stable co-localization of PKH and GFP in the bone marrow cells for more than 6 weeks was confirmed (data not shown).

## Real-time reverse transcription PCR (RT-PCR)

The eyes of laser-irradiated mice were enucleated at the selected times after the photocoagulation. Total RNA was extracted from the retina and choroidal sheets with the RNeasy mini kit (Qiagen, Hilden, Germany) and transcribed using the QuantiTect Reverse Transcription Kit (Qiagen). The cDNAs were amplified with QuantiTect SYBR Green PCR kit (Qiagen) with primer pairs (*Supplementary file 1*), and quantified using the LightCycler (Roche, Mannheim, Germany).

## Retinal microvascular endothelial cells and endothelial progenitor cells

Primary retinal vascular endothelial cells were collected from murine (C57-6065, Cell Biologics, Chicago, IL) and human retinas (ACBRI 181, Cell Systems, Kirkland, WA).

Primary retinal microvascular endothelial cells of *C57BL/6* mice were isolated from the retinal tissue of pathogen-free laboratory mice. The cells were negative for bacteria, yeast, fungi, and mycoplasma. Primary human retinal microvascular endothelial cells were also examined for absence of human immunodeficiency virus (HIV), hepatitis B virus (HBV), and hepatitis C virus (HCV) contaminations by serologic or PCR test (by CLIA Licensed Clinical Lab) and *Mycoplasma* spp. contaminations (ATCC method by CLIA Licensed Clinical Lab). They were propagated to confluence on gelatin-coated 96-well plates in Dulbecco's modified Eagle's medium (DMEM; Gibco, Grand Island, NY) supplemented with 10% fetal bovine serum, L-glutamine, endothelial cell growth supplement (Sigma, St. Louis, MO), heparin, and non-essential amino acids (Gibco).

To create endothelial progenitor cells (EPCs), isolated bone marrow cells of mice were plated on fibronectin-coated plates. The nonadherent cells were removed, and the attached cells were cultured for 2 weeks in DME/F-12 supplement with 15% FBS and recombinant GMCSF (10 ng/ml, Peprotec, Rocky Hill, NJ) (*Wang et al., 1998*). Colony forming units were formed and attached at 1 week. To confirm an endothelial cell lineage, the cells were stained with endothelial cell markers including CD31, VCAM-1, and von Willebrand factor. Briefly, cells plated on temperature-responsive dishes (CellSeed, Tokyo, Japan) were non-enzymatically dispersed and stained for FACS analysis.

## Tube formation assay of endothelial cells and microarray analysis

To examine the roles played by cytokines in angiogenesis, vascular endothelial cells were assayed for in vitro tube formation as described in detail (*DeCicco-Skinner et al., 2014*). Briefly, bone marrow-derived EPCs or retinal vascular endothelial cells were plated on Matrigel-coated plates with or without recombinant mouse IL-4 (R and D Systems) or recombinant human IL-4 (Peprotec, for human endothelial cells), and the presence of tube networks was quantified by digitization of the photographs by Photoshop (Adobe, San Jose, CA) after 24 hr.

The gene and the pathway associated with the tube formation were determined by microarray analysis of tube forming EPCs. EPCs derived from *wild type* or *Il4ra$^{-/-}$* mice were plated on Matrigel plates with or without IL-4 (10 ng/ml) to examine for tube formation. The total RNA was extracted using RNeasy mini kit (Qiagen) and analyzed using SurePrint G3 Mouse GE 8 × 60K Microarray (Agilent Technologies, Santa Clara, CA) (*Miyazaki et al., 2017*). The microarray data were analyzed using GeneSpring Software with setting of single color array and a fold change cut off of 3 and a p<0.05. A set of the IL-4-induced genes in the bone marrow-derived EPCs was analyzed to identify the canonical pathways and upstream regulators using Ingenuity Pathway Analysis software (IPA, Qiagen, accessed on 2020/4/5). The transcriptional networks of IL-4-stimulated EPCs were also

constructed using IPA and evaluated by the *P* value as likelihood that assembly of the genes in a network could be explained by random chance alone.

## Enzyme-linked immunosorbent assay (ELISA)

The supernatants of EPC were assayed with a commercial ELISA kit (ThermoFisher Scientific, Waltham MA). The levels of IL-4 and IL-13 in the aqueous humor samples were measured using commercial ELISA kit as described in detail (*Chono et al., 2018*; *Sasaki et al., 2012*).

## Statistical analyses

Data are presented as the means ± standard error of the means (SEMs). The significance of the differences was determined by two-tailed *t* tests, linear mixed-effects regression analysis, or ANOVA with post hoc tests. Logistic regression analysis was used to compute the odds ratios based on quintiles of each cytokine levels. A $p < 0.05$ was taken to be significant.

# Acknowledgements

We are grateful to Dr. Marc E Rothenberg who supplied the animal for the experiments using IL-13 receptor deficient mice. Dr. Tetsuya Ohbayashi kindly supported animal experiments conducted in Research Center for Bioscience and technology and animal care. Tottori Bio Frontier managed by Tottori prefecture kindly allowed access to confocal microscope, LSM730.

# Additional information

## Competing interests

Yoshitsugu Inoue: medical advisor of Senju Pharmaceutical Company, Japan. The other authors declare that no competing interests exist.

## Funding

| Funder | Grant reference number | Author |
|---|---|---|
| Japan Society for the Promotion of Science | JP16K15733 | Takashi Baba |
| Japan Society for the Promotion of Science | JP25670733 | Takashi Baba |

The funders had no role in study design, data collection and interpretation, or the decision to submit the work for publication.

## Author contributions

Takashi Baba, Conceptualization, Data curation, Formal analysis, Funding acquisition, Investigation, Writing - original draft, Writing - review and editing; Dai Miyazaki, Conceptualization, Data curation, Software, Formal analysis, Supervision, Writing - original draft, Project administration, Writing - review and editing; Kodai Inata, Ryu Uotani, Hitomi Miyake, Shin-ichi Sasaki, Yumiko Shimizu, Data curation, Formal analysis, Investigation; Yoshitsugu Inoue, Supervision, Funding acquisition, Project administration, Writing - review and editing; Kazuomi Nakamura, Resources, Formal analysis, Supervision, Methodology

## Author ORCIDs

Takashi Baba (iD) https://orcid.org/0000-0001-7318-9420

## Ethics

Human subjects: The study protocol was approved by the Ethics Committee of the Tottori University (protocol #: 2699), and the procedures used conformed to the tenets of the Declaration of Helsinki. An informed consent was obtained from all of the participants.

Animal experimentation: All mice were handled in accordance with the ARVO Statement for the Use of Animals in Ophthalmic and Vision Research and protocols approved by the Institutional Animal Care and Use Committee of Tottori University (protocol #: 12-Y-7, 13-Y-25, 16-Y-22, and 19-Y-50).

## Decision letter and Author response

Decision letter https://doi.org/10.7554/eLife.54257.sa1
Author response https://doi.org/10.7554/eLife.54257.sa2

## Additional files

### Supplementary files

• Supplementary file 1. Sequences of primer pairs used in quantitative reverse-transcription polymerase chain reaction.

• Transparent reporting form

### Data availability

All data generated or analysed during this study are included in the manuscript. Source data files have been provided for Figure 1, 2, 3, Figure 3—figure supplement 1, 2, Figure 4, Figure 4—figure supplement 1 and Figure 5.

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
