## [Decision Letter]

**Acceptance summary:**

The authors studied the role of IL-4 in choroidal neovascularization (CNV) in a mouse model. CNV is vision threatening in age related macular degeneration, commonly seen in the elderly population. They found that the IL-4/IL-4Rα axis contributed to CNV through communications with CCR2^+^ bone marrow cells which were recruited to CNV lesions. CCL2 enhanced recruitment of CCR2^+^ bone marrow cells. For CNV development, IL-4 in bone marrow cells is required, and IL-4 promotes CNV formation.

**Decision letter after peer review:**

Thank you for submitting your article "Role of IL-4 in bone marrow driven dysregulated angiogenesis and age-related macular degeneration" for consideration by *eLife*. Your article has been reviewed by three peer reviewers, one of whom is a member of our Board Reviewing Editors, and the evaluation has been overseen by Satyajit Rath as the Senior Editor. The following individual involved in review of your submission has agreed to reveal their identity: Ye Sun (Reviewer #2).

The reviewers have discussed the reviews with one another and the Reviewing Editor has drafted this decision to help you prepare a revised submission.

Summary:

This manuscript by Bab et al. identifies high levels of interleukin-4 in the aqueous of patients presenting with the neovascular form of AMD. From this observation, the authors go on to use a series of experimental techniques, both in vitro and in vivo, to suggest that IL-4 is rapidly expressed in experimental laser induced CNV lesions in a mouse model. This occurs in tandem with increases in CCL2 and CCR2 and the authors suggest that IL-4 is critical for the formation of CNV. They go on to show that systemic injection of IL-4 can exacerbate CNV lesions while inhibition of IL-4 can prevent CNV development. Finally, they show, using a series of bone marrow chimeric mice, that IL-4 is critical for CNV development and in its absence (systemically), CNV lesions cannot adequately form. While this study is of interest and the data presented are intriguing, there are numerous issues that need to be addressed that will clarify the datasets and make the study more robust.

Essential revisions:

1) In all figure legends n=8-9 eyes (not mice) per group were mentioned for each mouse strain/treatment. However, the description of the laser-induced CNV procedure was unclear, e.g. the number of laser shots per eye, as well as the number of mice per group was not stated. The laser-CNV model can have very large variations (PMID: 24136346; 26161975). Therefore, at least 8-10 mice per group, with 3-4 laser shots per eye have been recommended to obtain reliable results.

2) Please use scatter plots in all figures.

3) The authors claim that bone marrow-derived cells were recruited and differentiated into endothelial cells in CNV lesions (Figure 2E). However, isolectin is not a specific endothelial cell marker, as it also stains microglia (PMID: 23413818). As the authors have demonstrated, CCL2 is localized with microglia (Iba1 staining) in CNV lesions (Figure 1C). Therefore, the authors need to use more specific markers for endothelial cells, and for macrophage/microglia to further validate the type of cells that bone marrow-derived cells differentiate into. If it is macrophage/microglia cells, the authors need to re-validate the data in these cells instead of endothelial progenitor cells in Figure 3-4.

4) In Figure 2E and Figure 5B, the authors showed the co-staining of bone marrow marker and endothelial cell marker and concluded that bone marrow derived cells incorporated into CNV lesion. But those cells may just simply stick on top of the CNV lesion through cell fusion instead of incorporation into CNV endothelium. If the authors can provide Z-stack confocal images to show those cells in same location are also in the same layer, not on top of endothelial cells.

5) The authors examined how IL-4 affected the differentiation of vascular endothelial progenitor cells from bone marrow cells in Figure 3 by measuring the induction of angiogenesis related factors including CCL2, VEGF, VEGF receptors, angiopoietin-1, endothelial receptor, thrombin receptors, P-selectin, and vascular endothelial cadherin, and found that CCL2 and VEGFR1 were significantly induced upon IL-4 exposure. This experiment was done in cell culture and EPC markers were not used to confirm the cell lineage of EPC. Do IL-4 exposure changes the amounts/levels or timing of differentiation of vascular endothelial progenitor cells from bone marrow cells?

6) In Figure 1C, the authors showed the colocalization of IL-4 and Iba-1 and concluded that IL-4 was associated mainly with Iba-1-positive retinal microglial cells. Can the authors rule out the possibility of endothelial cells?

Images in Figure 1B really need a counterstain to show the demarcation of the lesions. For example, an isolectin stain or CD31 stain could be used to highlight the lesion independent of Il-4 or CD11b staining.

7) Why has a time course not been completed for Iba-1 and CCL2 staining? What does this staining look like at the times of peak IL4 expression? The authors state in the legend hat the Iba1 and CCL2 staining are in the retinal tissue. Was the retina left intact during these flat mounts or was it removed? Are these microglia adhered to the neovascular membrane or are they within the retina?

8) Figure 2: It would be beneficial to include similar time-points as those outlined in Figure 1A for each of the markers stained for in Figure 2E. As it stands, this only represents what appears to be occurring 2 weeks post laser injury. As this is a temporally developing lesion, it would be important to stain for these components at all other time points listed previously.

9) Figure 3: In the endothelial progenitors cells stimulated with IL-4 (Figure 3A), CCL2 is not increasing in a dose dependent manner as suggested by the authors, but rather is lower at 100 ng/ml than at 10 ng/ml. How is this explained?

While the authors show effects of IL-4 on endothelial CCL2 and VEGF mRNA expression, do they have data pertaining to protein expression from these similar experiments?

10) In Figure 3, it's strange that although IL-13 can induce CCL2 and VEGFR1 expression, inhibition of IL-13 has absolutely no effect on CNV formation as outlined in Figure 2. This should be elaborated upon in detail.

11) Figure 5: Images should be shown for each bone marrow chimera generated, not just for *IL-4^-/-^* mice.

12) Discussion: The authors need to address the apparent contradictory results between their study and the one they have cited in Wu et al., 2015. This study shows that systemic administration of IL-4 prevents CNV development whereas their own data suggests it augments it. This needs to be clarified with further discussion.

[Editors' note: further revisions were suggested prior to acceptance, as described below.]

Thank you for re-submitting your article "Role of IL-4 in bone marrow driven dysregulated angiogenesis and age-related macular degeneration" for consideration by *eLife*. Your revised article has been evaluated by a Reviewing Editor and Satyajit Rath as the Senior Editor. The Reviewing Editor has drafted this decision to help you prepare a revised submission.

Summary:

Choroidal neovascularization (CNV) in age-related macular degeneration (AMD) is a major cause of vision loss. CNV is associated with high local levels of IL-4. This study examined the role of IL-4 in CNV formation in a murine model. IL-4/IL-4 receptors (IL4Rs) controlled proangiogenic responses of bone marrow cells. CCR2^+^ bone marrow cells were recruited to very early CNV lesions. IL-4 rapidly induces CCL2, which enhances recruitment of CCR2^+^ bone marrow cells. For CNV development, IL-4 in bone marrow cells is required, and IL-4 promotes CNV formation. The IL-4/IL-4Rα axis contributes to pathological angiogenesis through communications with bone marrow cells leading to retinal degeneration.

Essential revisions:

1) The interactome in Figure 4—figure supplement 1 needs to be simplified as it is almost illegible in its complexity.

2) The authors still need to do some editing to tone down the conclusion that bone marrow-derived cells differentiated into endothelial cells as the data does not completely support this conclusion. Although the staining with endothelial cell CD31 lends some support, CD31 is expressed not only in vascular endothelial cells but also in T cells, B cells, dendritic cells (DCs), neutrophils, monocytes, and macrophage (PMID: 31732534). Please note this in the text.

3) In the text, the authors need to change their conclusion after adding CD31 staining in Figure 2: "In the CNV lesion, bone marrow-derived cells (green) were localized to isolectin-positive vascular endothelial cells indicating that the bone marrow-derived cells had differentiated into endothelial cells." The conclusion needs to be changed to "…CD31-positive…indicating that the bone marrow-derived cells may be able to differentiate into endothelial cells." and discuss the specificity of CD31 in endothelial cells.

---

## [Author Response]

Essential revisions:1) In all figure legends n=8-9 eyes (not mice) per group were mentioned for each mouse strain/treatment. However, the description of the laser-induced CNV procedure was unclear, e.g. the number of laser shots per eye, as well as the number of mice per group was not stated. The laser-CNV model can have very large variations (PMID: 24136346; 26161975). Therefore, at least 8-10 mice per group, with 3-4 laser shots per eye have been recommended to obtain reliable results.

Because there were three laser irradiation sites/eye, the nested ANOVA test was used to compare the size of the CNV area between groups. We showed each CNV area/spot with dots in all figures.

Our experiments were conducted using at least 8-10 eyes (8-12 mice), and 3 laser spots were made in each eye. The descriptions were changed to spots and eyes. Because the spots were nested in the eyes, nested ANOVA analysis was used.

Subsection “Induction of choroidal neovascularization”: “Three laser spots were created in each eye.”

Figure 2 legend: “Nested ANOVA with post hoc test.”

Figure 5 legend: “Nested ANOVA with post hoc test.”

Some of the IL-4 chimeric mice did not survive through the experimental period. This suggested that IL-4 plays a critical role in the survival of bone marrow cells. The number of dead mice is shown in the figure legends.

Figure 5 legend: “Six out of 10 *Il4^-/-^* bone marrow chimeric mice in each group did not survive through procedures, and/or euthanized.”

2) Please use scatter plots in all figures.

Plots were changed to scatter graphs.

3) The authors claim that bone marrow-derived cells were recruited and differentiated into endothelial cells in CNV lesions (Figure 2E). However, isolectin is not a specific endothelial cell marker, as it also stains microglia (PMID: 23413818). As the authors have demonstrated, CCL2 is localized with microglia (Iba1 staining) in CNV lesions (Figure 1C). Therefore, the authors need to use more specific markers for endothelial cells, and for macrophage/microglia to further validate the type of cells that bone marrow-derived cells differentiate into. If it is macrophage/microglia cells, the authors need to re-validate the data in these cells instead of endothelial progenitor cells in Figure 3-4.

The purpose of our study was to determine how IL-4 contributed to bone marrow-mediated CNV formation. To understand the kinetics and role of bone marrow cells, we conducted another time series analysis after bone marrow transfer. For this series of analyses, CD31 staining was used to identify endothelial cells (Figure 2—figure supplement 1).

We showed that the initial recruitment of bone marrow cells peaked at 3 days after the irradiation. Bone marrow cells in this priming phase are CD11b^+^, and they secrete CCL2. This population are not positive for iba1 and are morphological distinct from the accumulated microglial cells. This suggested that these cells will enhance the recruitment using CCL2 signals.

After the priming phase subsides, CD31-positive bone marrow cells are observed in the center of the CNVs. Because these cells secrete IL-4, endothelial cells, either bone marrow derived or not, are stimulated. Our in vitro tube formation experiments using endothelial cells as well as bone marrow-derived cells supports this stimulatory effect of IL-4.

“Recruitment of GFP^+^ bone marrow derived cells (green) peaked at 3 days after irradiation. Thus, bone marrow-derived cell recruitment also contributed to the inductive phase process. Bone marrow-derived cells in this phase were CD11b^+^ lineage, and they were positive for CCL2. These bone marrow cells did not express iba1 and were morphologically distinct from microglial cells. This suggested that these cells will amplify the recruitment of CCR2^+^ lineage cells (Figure 2—figure supplement 1).”

4) In Figure 2E and Figure 5B, the authors showed the co-staining of bone marrow marker and endothelial cell marker and concluded that bone marrow derived cells incorporated into CNV lesion. But those cells may just simply stick on top of the CNV lesion through cell fusion instead of incorporation into CNV endothelium. If the authors can provide Z-stack confocal images to show those cells in same location are also in the same layer, not on top of endothelial cells.

We conducted 3D analysis on bone marrow chimeric mice (Figure 2—video 1). In the priming phase, the bone marrow cells were seen attached to the budding CNV. This phase appeared as the amplification phase. CD31 positive bone marrow cells with endothelial morphology were seen after 7 days.

“Seven days after irradiation, bone marrow-derived cells were incorporated into CD31^+^ endothelial cells, and they had endothelial cell morphology (Figure 2—video 1).”

5) The authors examined how IL-4 affected the differentiation of vascular endothelial progenitor cells from bone marrow cells in Figure 3 by measuring the induction of angiogenesis related factors including CCL2, VEGF, VEGF receptors, angiopoietin-1, endothelial receptor, thrombin receptors, P-selectin, and vascular endothelial cadherin, and found that CCL2 and VEGFR1 were significantly induced upon IL-4 exposure. This experiment was done in cell culture and EPC markers were not used to confirm the cell lineage of EPC. Do IL-4 exposure changes the amounts/levels or timing of differentiation of vascular endothelial progenitor cells from bone marrow cells?

The differentiation protocol was based on previous studies (Wang et al., 1998), and was also reviewed by Calzi et al. Microvascular Res 2010 (doi:10.1016/j.mvr.2010.02.011).

Currently, the described protocols do not require the addition of IL-4. We used IL-4 after differentiation. We assumed that the intrinsic IL-4 was sufficient for full differentiation or IL-4 will activate endothelial cells.

6) In Figure 1C, the authors showed the colocalization of IL-4 and Iba-1 and concluded that IL-4 was associated mainly with Iba-1-positive retinal microglial cells. Can the authors rule out the possibility of endothelial cells?Images in Figure 1B really need a counterstain to show the demarcation of the lesions. For example, an isolectin stain or CD31 stain could be used to highlight the lesion independent of Il-4 or CD11b staining.

Microglial cells were co-stained with Iba-1 and CD11b (not IL-4) and were located on the surface of the CNVs beneath the retinal vessels 1 to 3 days after laser treatment. (Figure 1—video 1).

“The microglial cells migrated to surface of the CNV (Figure 1—video 1).”

“Figure 1—video 1

Localization of bone marrow-derived cells and microglial cells in the retinal tissue of GFP bone marrow chimeric mouse in 3D rendering at 3 days after laser irradiation.

Bone marrow-derived cells (green) are located in the subretinal layer around the laser irradiation lesion 3 days after laser irradiation. Bone marrow-derived cells (green) expressing CD11b (cyan) was spatially and morphologically distinct from accumulated microglial cells (arrows). CD11b (cyan) positive microglial cells migrated on CD31 (red) positive retinal and choroidal vessels, and are observed on the surface of choroidal neovascularization. Scale 50 μm.”

7) Why has a time course not been completed for Iba-1 and CCL2 staining? What does this staining look like at the times of peak IL4 expression? The authors state in the legend hat the Iba1 and CCL2 staining are in the retinal tissue. Was the retina left intact during these flat mounts or was it removed? Are these microglia adhered to the neovascular membrane or are they within the retina?

We showed the time course for Iba-1 and CCL2 staining in retinal flat mounts (Figure 1—figure supplement 1). Lesional IL-4 peaks at 3 days after the laser irradiation. IL-4 expressing bone marrow cells were mainly CD11b^+^, and they had monocytic morphology.

“The kinetics of Iba1, CCL2, and CD11b positive cells after laser exposure was consistent with that of the mRNA induction (Figure 1—figure supplement 1).”

8) Figure 2: It would be beneficial to include similar time-points as those outlined in Figure 1A for each of the markers stained for in Figure 2E. As it stands, this only represents what appears to be occurring 2 weeks post laser injury. As this is a temporally developing lesion, it would be important to stain for these components at all other time points listed previously.

We prepared kinetics panel staining as shown in Figure 2—figure supplement 1. IL-4 secretion is generally a very small in amount. Using immunohistochemistry, IL-4 was not obviously expressed during the one to three days and was not shown.

Recruitment of GFP^+^ bone marrow derived cells (green) peaked at 3 days after irradiation. Thus, bone marrow-derived cell recruitment also contributed to the inductive phase process. Bone marrow-derived cells in this phase were CD11b^+^ lineage, and they were positive for CCL2. These bone marrow cells did not express iba1 and were morphologically distinct from microglial cells. This suggested that these cells will amplify the recruitment of CCR2^+^ lineage cells (Figure 2—figure supplement 1).

9) Figure 3: In the endothelial progenitors cells stimulated with IL-4 (Figure 3A), CCL2 is not increasing in a dose dependent manner as suggested by the authors, but rather is lower at 100 ng/ml than at 10 ng/ml. How is this explained?

IL-4 of 100 ng/ml is a high in concentration for in vivo conditions. The optimal concentration of IL-4 is within 10 ng/ml at most. The IL-4 concentration of 100 ng/ml was used to confirm the blocking effect of anti-IL-4Rα antibody.

While the authors show effects of IL-4 on endothelial CCL2 and VEGF mRNA expression, do they have data pertaining to protein expression from these similar experiments?

We confirmed that the protein expression of endothelial CCL2 and VEGFR-1 stimulated 1 ng/mL or 10 ng/mL by ELISA (Figure 3—figure supplement 2).

“An upregulation of the translation of CCL2 and VEGFR-1 in EPCs was confirmed by ELISA. IL-4 exposed EPCs had a significant increase in the secretion of CCL2 (*P* = 0.000) and VEGFR-1 (*P* = 0.000) after 24 h exposure to IL-4 (Figure 3—figure supplement 2).”

10) In Figure 3, it's strange that although IL-13 can induce CCL2 and VEGFR1 expression, inhibition of IL-13 has absolutely no effect on CNV formation as outlined in Figure 2. This should be elaborated upon in detail.

CCL2 and VEGFR1 are representatives of important genes for the angiogenesis process. However, angiogenesis requires numerous factors, which operate in coordination kinetically and spatially. We assume that the IL-13/IL-13Rα axis may not induce some other essential factors.

11) Figure 5: Images should be shown for each bone marrow chimera generated, not just for IL-4^-/-^ mice.

We prepared panels of bone marrow cells and CNV as Figure 5—figure supplement 1.

Figure 5—figure supplement 1

Localization of bone marrow-derived cells (green) and endothelial cells (red) in the retinal tissue of *wild type* and *Ilra^-/-^* bone marrow chimeric mice on *Il4ra^-/-^* background and wild type, *Il4^-/-^* and *Il4ra^-/-^* bone marrow chimeric mice on *wild type* background. Endothelial cells in the CNV were labeled with IB4 (red). Reduced sized CNV contained wild type bone-derived cells. Scale 10 μm.

12) Discussion: The authors need to address the apparent contradictory results between their study and the one they have cited in Wu et al., 2015. This study shows that systemic administration of IL-4 prevents CNV development whereas their own data suggests it augments it. This needs to be clarified with further discussion.

The direct suppressive effect of IL-4 on CNV development is shown using vitreous injection of IL-4 by Wu et al. However, the concentration used by Wu et al. was extremely high (600 ng/ml). This extremely high concentration is not necessary under in vivo conditions, and IL-4 is not expressed in such high concentration in the eye. This level is highly toxic and can damage endothelial cells or recruited cells. Other experiments showing the suppressive effect of IL-4 was conducted using in vitro differentiated macrophages presumably reflecting the stimulating differentiation into regulatory phenotype by IL-4.

“Wu et al. also showed a contradictory role of IL-4 for CNV formation (Wu et al., 2015). This was shown using vitreous injection of IL-4 at very high concentration (600 ng/ml). This may cause toxic damage to endothelial cells or recruited cells which may not reflect physiological role of IL-4.”

[Editors' note: further revisions were suggested prior to acceptance, as described below.]

Essential revisions:1) The interactome in Figure 4—figure supplement 1 needs to be simplified as it is almost illegible in its complexity.

Figure 4—figure supplement 1B

The primary networks were confined to the top 3 major networks. Fewer genes and molecules are shown in the merged network. To improve the understanding, the layout of the genes was changed to a hierarchical view.

Subsection “Tube formation assay of endothelial cells and microarray analysis”: “... (IPA, Qiagen, accessed on 2020/4/5).”

Legend to Figure 4—figure supplement 1:

“A: IL-4Rα-mediated transcriptional networks associated with angiogenesis in bone marrow-derived EPCs after IL-4 treatment. Z score = 2.781, P = 1.8×10^-4^

B: IL-4Rα-mediated transcriptional networks of bone marrow-derived EPCs. The top 3 highest significant networks (P <1×10^-27^) were merged and are shown in a hierarchical layout. Amyloid-β A4 protein (App) and cellular tumor antigen p53 (Tp53) are shown in the top and bottom positions. Red indicates an upregulation and green indicates a downregulation. The IPA was accessed on 2020/4/5.”

2) The authors still need to do some editing to tone down the conclusion that bone marrow-derived cells differentiated into endothelial cells as the data does not completely support this conclusion. Although the staining with endothelial cell CD31 lends some support, CD31 is expressed not only in vascular endothelial cells but also in T cells, B cells, dendritic cells (DCs), neutrophils, monocytes, and macrophage (PMID: 31732534). Please note this in the text.

Definitive proof of differentiation of bone marrow-derived cells requires further meticulous experimentations. Our major focus was to determine the contribution of bone marrow cells and IL-4 to CNV formation. Although our data are consistent with the bone marrow cells differentiation into endothelial cells, the context was changed to suggest other possibility that bone marrow cells may behave as non-endothelial lineage.

“To identify an endothelial lineage, we used CD31 or isolectin staining, because CD31 is highly expressed on endothelial cells and is commonly used as an endothelial cell marker. However, CD31 can also be expressed in other lineage cells including T cells, B cells, dendritic cells (DCs) (Clement et al., 2014), neutrophils, monocytes, and macrophage (Merchand-Reyes et al., 2019).”

“Our data support the idea that bone marrow-derived cells may be able to differentiate into endothelial cells. However in the CNV lesion, whether endothelial differentiation is complete or bone marrow cells serve as immature or a different lineage is uncertain. Importantly, bone marrow-derived cells do play pivotal roles in the CNV formation.”

3) In the text, the authors need to change their conclusion after adding CD31 staining in Figure 2. "In the CNV lesion, bone marrow-derived cells (green) were localized to isolectin-positive vascular endothelial cells indicating that the bone marrow-derived cells had differentiated into endothelial cells." The conclusion need to be changed to "…CD31-positive…indicating that the bone marrow-derived cells may be able to differentiate into endothelial cells." and discuss the specificity of CD31 in endothelial cells.

Immunohistochemistry showed a colocalization of CD31 and bone marrow cells. However, a co-expression of CD31 does not completely exclude the possibility that bone marrow cells may have differentiated into another lineage or remain relatively immature. The description in the Results section was revised to that shown below to suggest this other possibility. We also added text that non-endothelial cells may express CD31 in the Discussion section.

“... bone marrow-derived cells were incorporated into the structures formed by CD31^+^ endothelial cells” (Figure 2—video 1).

“In the CNV lesion, bone marrow-derived cells (green) were localized to the structures formed by isolectin-positive vascular endothelial cells, although this may not indicate complete differentiation into endothelial lineage. However, the co-localization of the marrow-derived cells to structures formed by CD31^+^ endothelial cells indicates that the bone marrow-derived cells might be able to differentiate into endothelial cells.”

“Alternatively, bone marrow cells will also differentiate into a non-endothelial cell lineage serving as provider of CNV forming signals.”

“... the IL-4R^+^ bone marrow-derived cells were incorporated into the CNV presumably as late EPCs together with the resident cell-derived endothelial cells.”